# Phase-Dependent Differential In Vitro and Ex Vivo Susceptibility of *Aspergillus flavus* and *Fusarium keratoplasticum* to Azole Antifungals

**DOI:** 10.3390/jof9100966

**Published:** 2023-09-26

**Authors:** Darby Roberts, Jacklyn Salmon, Marc A. Cubeta, Brian C. Gilger

**Affiliations:** 1Department of Clinical Sciences, College of Veterinary Medicine, North Carolina State University, Raleigh, NC 27606, USA; drobert6@ncsu.edu (D.R.);; 2Department of Entomology and Plant Pathology, Center for Integrated Fungal Research, North Carolina State University, Raleigh, NC 27606, USA; macubeta@ncsu.edu

**Keywords:** *Aspergillus*, *Fusarium*, fungal keratitis, antifungal susceptibility, MIC, azole

## Abstract

Fungal keratitis (FK) is an invasive infection of the cornea primarily associated with *Aspergillus* and *Fusarium* species. FK is treated empirically with a limited selection of topical antifungals with varying levels of success. Though clinical infections are typically characterized by a dense network of mature mycelium, traditional models used to test antifungal susceptibility of FK isolates exclusively evaluate susceptibility in fungal cultures derived from asexual spores known as conidia. The purpose of this study was to characterize differences in fungal response when topical antifungal treatment is initiated at progressive phases of fungal development. We compared the efficacy of voriconazole and luliconazole against in vitro cultures of *A. flavus* and *F. keratoplasticum* at 0, 24, and 48 h of fungal development. A porcine cadaver corneal model was used to compare antifungal efficacy of voriconazole and luliconazole in ex vivo tissue cultures of *A. flavus* and *F. keratoplasticum* at 0, 24, and 48 h of fungal development. Our results demonstrate phase-dependent susceptibility of both *A. flavus* and *F. keratoplasticum* to both azoles in vitro as well as ex vivo. We conclude that traditional antifungal susceptibility testing with conidial suspensions does not correlate with fungal susceptibility in cultures of a more advanced developmental phase. A revised method of antifungal susceptibility testing that evaluates hyphal susceptibility may better predict fungal response in the clinical setting where treatment is often delayed until days after the initial insult.

## 1. Introduction

Fungal keratitis (FK), a form of infectious keratitis primarily associated with species of *Aspergillus, Candida*, or *Fusarium*, is reported to be under-recognized and under-prioritized [1,2,3]. Modern estimates report a global incidence of 1.5 million cases per year, with at least 60% of cases resulting in loss of vision in the affected eye despite the best available treatment [1,2,4]. Yet, in a 2021 report by the Lancet Commission entitled “Global Eye Health: vision beyond 2020”, fungal keratitis was not mentioned [5]. A lack of recognition may be due in part to the relative exclusivity of the disease to human populations of low-income, tropical regions [2,6]. This puts fungal keratitis at low priority for medical researchers in the United States and other developed countries. 

For veterinary researchers in the U.S., however, fungal keratitis is of high priority due to the prevalence of the disease in equine populations. In horses, fungal keratitis is caused by the same fungal species implicated in human infection and has the same clinical symptoms [7,8]. Further, veterinarians face many of the same obstacles to treatment as physicians. These include delayed clinical presentation and diagnosis, difficulty selecting and/or obtaining appropriate drugs, increasing fungal resistance to available drugs, and poor patient compliance due to the necessary high frequency of treatments [9,10]. As a result of these challenges, medical management with modern antifungals is often not effective [9]. Therefore, there is an urgent need for research of novel methods of treatment, including but not limited to broad-spectrum antifungal drugs, as well as methods and models that will reliably assess both in vitro susceptibility and practical clinical efficacy. 

The current gold standard method for evaluating antimicrobial susceptibility is the broth microdilution minimum inhibitory concentration (MIC) assay [11]. These results, however, are notoriously challenging to interpret, particularly in the absence of established drug breakpoints for most *Aspergillus* and *Fusarium* species [12]. In vitro resistance in the face of clinical efficacy, and vice versa, are both regularly reported [8,13,14]. A driving factor of this incongruity is that traditional MIC assays exclusively test antifungals against a suspension of asexual fungal spores called conidia [11]. However, FK is often characterized by the presence of mature, hyphal fungi, often encased in a biofilm [15]. Biofilm formation has been reported to contribute to antifungal resistance in *Fusarium* FK infections as well as reduced antifungal susceptibility in cultures of *Fusarium*, *Candida*, and *Aspergillus* [16,17,18]. Therefore, to make a practical prediction regarding clinical efficacy of an antifungal, biofilm susceptibility must be considered alongside already reported conidial susceptibility. 

The purpose of this study was to characterize differences in response of FK associated fungal species when antifungal treatment with the reported minimum inhibitory concentration of two azole antifungals is initiated at sequential stages of fungal development. To accomplish this, we investigated two methods of evaluating antifungal susceptibility against fungal conidia and hyphal growth stages. To evaluate the response and susceptibility of conidia to two antifungals in vitro, an adaptation of the Clinical and Laboratory Standards Institute (CLSI) reference “Broth Dilution Antifungal Susceptibility Testing of Filamentous Fungi” fungi (M38, 3rd ed.) was used [19]. Modifications to the standard in vitro protocol included the addition of two sample groups incubated for either 24 or 48 h prior to introducing an antifungal drug. The second method evaluated susceptibility using an ex vivo porcine corneal cadaver model of intrastromal infection. In this model, the antifungal was introduced to sample groups at either the conidial stage, after 24 h or 48 h of incubation. For both methods, fungal growth inhibition was compared between sample groups treated at the conidial stage versus those treated at mixed hyphal stages. The antifungal drugs selected for this study were voriconazole, a second generation triazole, and luliconazole, an imidazole [20,21]. As the 24 and 48-h sample groups were designed to target fungal hyphae after spore germination and hyphal germ tube development, time-lapse confocal imaging was performed on *Aspergillus* and *Fusarium* in in vitro and ex vivo experiments to identify the incubation period necessary for conidial germination of both species.

## 2. Materials and Methods

### 2.1. Fungal Inoculum Preparation

#### 2.1.1. Selection and Genetic Characterization

Two fungal species were selected for their prevalence in clinical keratitis cases, *Aspergillus flavus* and *Fusarium keratoplasticum*. Clinical isolates were originally collected from equine patients with fungal keratitis. Complete preservation, identification, and genomic sequencing were performed previously and are described here in brief [7]. After initial culture sample evaluation and identification, isolates were further cultured for purification, at such time single conidial-derived stocks were prepared and stored frozen at −80 °C in 2 mL cryogenic vials until the time of use. Multi-locus sequence typing (MLST) was performed to confirm species identification and identify evolutionary lineage and/or species haplotype [7]. The *A. flavus* isolate used for this study was identified as MLST designation AF9 lineage subgroup IC, while the *F. keratoplasticum* isolate used was identified as MLST designation FK1 haplotype 2u [7]. 

#### 2.1.2. Propagation and Collection

Fresh cultures were prepared from frozen conidial stocks by flash-thawing cryovials in a 45 °C water bath until few ice crystals remain, briefly vortexing, and pipetting 100 μL of conidial suspension onto a 100 mm diameter plastic petri plate containing Potato Dextrose Agar (PDA, Difco, Franklin Lakes, NJ, USA) followed by spreading with a sterile L-shaped cell spreader. Plates were incubated at 33 °C, 5% CO_2_ for 3–5 d. At the time of collection, each plate was flooded with 5 mL potato dextrose broth prepared at a 50% dilution (PDB50, Difco, Franklin Lakes, NJ, USA). The agar surface was scraped using a sterile rubber cell scraper and the resulting suspension filtered through sterile cheesecloth lining a funnel. The suspension was diluted 1:10 with Potato Dextrose Broth (PDB50) and the concentration was determined by counting conidia with a hemocytometer. 

### 2.2. Germination Characterization

Widefield fluorescence imaging was performed using a Leica DM6000M microscope (Leica Microsystems, Deerfield, IL, USA) enclosed in a DM 6000 CS incubator (PeCon, Erbach, Germany) maintained at 33 °C, 5% CO_2_. To determine the time to conidial germination in vitro, 100 μL of a 200 spore/mL conidial suspension of each isolate was prepared as described (Section 2.4) and spread onto a 60 mm diameter PDA plate. To determine the time to germination ex vivo, corneas were inoculated as described (Section 2.5) and placed into a 60 mm tissue culture-treated dish (Corning Inc., Corning, NY, USA) with 4–6 mL DMEM culture medium. Time-lapse images were acquired every 30 min over 40 h (Andor Clara Interline CCD Camera; Oxford Instruments, Morrisville, NC, USA) in the form of a Z-stack of images. Representative images were chosen from each time point and stitched into video format using LAS X software (Leica Microsystems, Version 1.4.4, Deerfield, IL, USA). 

### 2.3. Antifungal Drug Preparation

The antifungal drugs selected for this study were voriconazole, a second generation triazole commonly prescribed for empirical treatment of fungal keratitis, and luliconazole, an imidazole most well-known for treatment of dermatophytosis [20,21]. For all experiments, voriconazole was prepared at a concentration equal to the MIC for each isolate as previously determined: 0.5 μg/mL for *A. flavus* experiments and 8 μg/mL for *F. keratoplasticum* experiments. Luliconazole was prepared at a concentration 2-log higher than the MIC due to the inconsistent solubility at lower concentrations [22]: 0.1 μg/mL for *A. flavus* experiments and 0.2 μg/mL for *F. keratoplasticum* experiments. A voriconazole stock solution was prepared using dry analytical grade drug (Ref: 32483, Lot: BCBS3721V, Sigma-Aldrich, St. Louis, MO, USA) dissolved in dimethyl sulfoxide (DMSO, Sigma-Aldrich, St. Louis, MO, USA). Luliconazole 1% suspension was prepared using dry drug (Millipore Sigma, Burlington, MA, USA) and further diluted with sterile deionized water. For all ex vivo assays, antifungal culture medium was prepared by adding antifungal drug dilution to Dulbecco’s Modified Eagle Medium (DMEM, Sigma-Aldrich, St. Louis, MO, USA) containing 10,000 U/mL penicillin and 10,000 μg/mL streptomycin (Thermo Fischer Scientific, Waltham, MA, USA).

### 2.4. In Vitro Antifungal Susceptibility Assay

A modified version of the CLSI reference standard M38 broth microdilution protocol was used, as previously described by our laboratory [19,22]. *A. flavus* and *F. keratoplasticum* conidial suspensions were prepared as described above and diluted to a concentration of either 200,000 or 40,000 conidia/mL, respectively. Each sample well of a 96-well plate received 50 μL conidial suspension plus 150 μL PDB50. Each treatment well also received 1 μL antifungal drug at either 0, 24, or 48 post-incubation. Negative control wells received 200 μL PDB50 only, while positive control wells received 50 μL conidial suspension plus 150 μL PDB50. All plates were incubated at 33 °C, 5% CO_2_ for 72 h. Wells were visually examined at 72 h using a magnifying reading mirror (Figure 1). Wells that visually displayed 90% or greater reduction in growth in comparison to the positive control were considered susceptible. Absorbance at 72 h was evaluated at 490 nm (Sunrise Microplate Reader, Tecan Group Ltd., Männedorf, Switzerland), according to methods of Rodrigues et al. [23].

### 2.5. Ex Vivo Intrastromal Infection Model

#### 2.5.1. Intrastromal Injection

Porcine cadaver eyes (Animal Technologies, Inc., Tyler, TX, USA) were shipped on ice and received within 24 h of ocular collection. Prior to intrastromal injection, each eye was disinfected by dropping 1% bleach (NaOCl) onto the cornea for 10 s, submerging the cornea in 1% betadine for 2 min, and rinsing with sterile Phosphate Buffered Saline (PBS, pH = 7.4). The disinfected eye was placed on a Mandell eye mount and intraocular pressure was adjusted to 15–20 mmHg to ensure repeatable depth of injection. A custom-made 600 mm, 34G microneedle was used to inject 25 μL of fungal conidial suspension into the center of the corneal stoma. Corneas were excised and rinsed with 1% NaOCl followed by PBS to remove residual conidial suspension solution from the surface. Each excised cornea was placed into a separate well of a 6-well tissue culture-treated plate (Corning Inc., Corning, NY, USA) containing 6–8 mL of culture medium. Plates were incubated at 33 °C, 5% CO_2_ for 72 h, with a medium change every 24 h.

#### 2.5.2. Antifungal Drug Treatment

Treatment groups were divided by fungal species, antifungal drug treatment, and time at which treatment was initiated post-inoculation. Culture medium was prepared with Dulbecco’s Modified Eagle Medium (DMEM, Sigma-Aldrich, St. Louis, MO, USA) containing 10,000 U/mL penicillin and 10,000 μg/mL streptomycin (Thermo Fischer Scientific, Waltham, MA, USA); antifungal culture medium was prepared via the addition of antifungal drug dilution, prepared as described above. For each fungus/drug combination, culture medium was replaced by antifungal culture either immediately post-injection, or 24 h or 48 h post-incubation. Three corneas (n = 3) were used for each treatment group. A control group of 3 corneas per isolate received culture medium without the antifungal agent post inoculation.

#### 2.5.3. Imaging and Growth Measurement

Corneas were retroilluminated and photographed (Nikon D200, AF-S FX Micro NIKKOR 105 mm 1:2.8G ED Lens; Nikon Corporation, Tokyo, Japan) every 12 h for 72 h following intrastromal injection. The camera focus distance was maintained at 0.33 m to ensure a controlled distance from corneal surface to lens. A light source with a diameter equivalent to the diameter of the excised corneas was used to align each cornea within the margins of the light source and provide consistent levels of incident light across images. The ImageJ (version 1.52k) “freehand selection” tool was used to quantify the area of radial fungal growth from the injection site by outlining the visible hyphal mass to attain a pixel count within the gated area). Percent corneal coverage by radial fungal growth was determined by comparing the area of the cornea on imaging obscured by intrastromal fungal growth to the total area of the cornea as measured by pixel count. This calculation was completed for each time point over 72 h.

### 2.6. Statistical Analysis

Associations among pixel counts of fungal growth or absorbance were evaluated using Analysis of Variance (ANOVA) and Tukey’s post hoc analysis for multiple comparisons. Differences were considered significant at *p* ≤ 0.05 and all probabilities and results were calculated using computerized statistical software (JMP^®^ Pro, v. 15.2; SAS Inc., Cary, NC, USA). 

## 3. Results

### 3.1. Characterization of Conidial Germination

Germination was considered to have occurred when greater than 90% of conidia in culture or ex vivo had formed germ tubes, which were characterized as having half the width and at least 2 times the length of the spore from which it derived (Appendix A). Time-lapse imaging of *A. flavus* conidia in vitro revealed germination occurred by 10 h of incubation. In vitro germination of *F. keratoplasticum* conidia was observed by 12 h of incubation. Ex vivo corneal germination of *A. flavus* conidia occurred by 16 h (Appendix A) and 17 h for conidia of *F. keratoplasticum.*

### 3.2. In Vitro Antifungal Susceptibility 

The goal of this assay was to compare in vitro susceptibility of ungerminated conidia versus a mixed culture of conidia, germlings, and hyphae. Our laboratory has previously determined the MIC of these isolates used in this study to VOR and LUL using a traditional CLSI Broth Microdilution technique [22]. To accomplish our goal, we prepared in vitro fungal cultures identical to those used in previous MIC assays and treated the cultures with the MIC dose of VOR or 2log the MIC dose of LUL after either 0, 24, or 48 h incubation (Figure 1).

Results were similar to our previous studies [22] when repeated with both *A. flavus* and *F. keratoplasticum.* Only the wells which received antifungal immediately post-plating (0 h Incubation), or after 24 h in the case of *F. keratoplasticum*, demonstrated fungal susceptibility to either drug, as determined by visual examination. Further, there was a significant reduction in absorbance compared to the positive control in the 0 h incubation group treated with either drug, as well as in the 24 h incubation group treated with VOR with both *A. flavus* and *F. keratoplasticum* (Figure 2). 

These results demonstrate that the MIC, of either voriconazole or luliconazole, as determined by a conidial suspension model does not correlate with significant inhibition in a suspension of mixed conidia, germlings, and hyphae. The results obtained from traditional CLSI antifungal susceptibility testing, as might be performed as part of an in-house diagnostic routine, should not be extrapolated to predict clinical efficacy. 

### 3.3. Ex Vivo Antifungal Susceptibility

We compared the growth of intrastromal fungi in a corneal ex vivo model following treatment with an antifungal drug after 0, 24, or 48 h of incubation (Figure 3). This design was intended to simulate clinical experience and the need to target various stages of fungal growth during fungal keratitis.

When performed in corneas inoculated with *A. flavus*, treating with either VOR or LUL after 0 h incubation resulted in >99% reduction in growth compared to the untreated control group. Treatment with either drug after 24 h of incubation resulted in >96% reduction in growth. However, treatment after 48 h of incubation resulted in only 25% reduction in growth in VOR-treated corneas and 0% reduction in growth in LUL-treated corneas (Figure 4A). 

Similarly, in corneas inoculated with *F. keratoplasticum*, treatment with either VOR or LUL after 0 h incubation resulted in >96% reduction in growth compared to the control group. Treatment after 24 h incubation resulted in 61–66% reduction in growth in LUL- or VOR-treated corneas, respectively. Finally, treatment after 48 h incubation resulted in <40% reduction in growth in VOR-treated corneas and <20% reduction in growth in LUL-treated corneas (Figure 4B). 

Delaying the onset of treatment following a corneal microconidial inoculation results in a reduced ability of the antifungal to inhibit further fungal growth. These results are similar to that observed in clinical FK patients where the efficacy of antifungal therapies is poor [3] and thus highlight the innate antifungal resistance of mature mycelial organisms compared to their conidial counterparts.

## 4. Discussion

Conventional antifungal susceptibility testing (AST), by either CLSI or EUCAST reference standards, requires fungal susceptibility to be evaluated from the conidial pre-germination stage. However, clinical manifestations of filamentous fungal infection are typically characterized by the destructive presence of hyphae with or without an encapsulating biofilm [16,24,25]. To be practically useful, antifungal susceptibility results need to offer clinicians information on drug efficacy across stages of fungal development. This has long since been recognized; some of the earliest attempts at establishing antifungal susceptibility testing for filamentous fungi focused on standardizing hyphal inocula and comparing results to MICs obtained from conidia [26,27]. While it was recognized then that MICs obtained from hyphal inocula tended to be several times higher, difficulties establishing a reliable technique for obtaining and quantifying a pure hyphal inocula ultimately discouraged the pursuit of hyphal MICs, and the reference standard became to use pure conidial inocula [11]. As a result, treatment decisions in cases of many fungal diseases must be made without critical information about the susceptibility of more advanced morphologies. In the case of fungal keratitis, medical treatment failure is reported in up to 30% of cases [14]. 

In this study, we designed and investigated a revised in vitro AST protocol based on the CLSI “Broth Dilution Antifungal Susceptibility Testing of Filamentous Fungi” reference standard to complement previous studies using this experimental approach, which allows for comparison between antifungal susceptibility in fungal colonies treated with an antifungal agent prior to and following conidial germination. We first confirmed via microscopy that complete germination occurred in vitro within 24 h of incubation for *A. flavus* and *F. keratoplasticum*. We then performed AST with conidial suspensions which were allowed to incubate for 0, 24, or 48 h before introduction of the antifungal agent. The 24 h incubation group was designed to mimic conditions of early-stage confluent hyphal growth after conidial germination while the 48 h incubation group was designed to model late-presenting fungal infection typical of clinical experience. We found that only the 0 h incubation groups of either *A. flavus* or *F. keratoplasticum* were susceptible to the MIC of both voriconazole and luliconazole as determined by complete inhibition of visual growth. 

We further adapted our in vitro protocol to an ex vivo model of FK to investigate fungal response to treatment initiated by either pre- or post-conidial germination in a biologically relevant environment. Our data demonstrates that treating corneas infected with *A. flavus* in the ex vivo model with voriconazole (0.5 ug/mL) or luliconazole (0.1 ug/mL) after 24 h incubation resulted in >96% reduction in growth compared to the untreated controls. However, when treatment was initiated at 48 h post-inoculation, reduction in growth was limited to 25% or less. Treating corneas infected with *F. keratoplasticum* in our ex vivo model with either voriconazole (8 ug/mL) or luliconazole (0.2 ug/mL) after 24 h of incubation resulted in 61–66% reduction in growth, while treating after 48 h of incubation limited reduction in growth to <40% for voriconazole and <20% for luliconazole. 

These results demonstrate a reduced response to azole treatment by fungal hyphae compared to conidia both in vitro and in a relevant ex vivo corneal infection model. This information is critical as clinical antifungal dosage recommendations are determined by MICs established against conidial suspensions, even though clinical infections are characterized by hyphae and mycelium which, as demonstrated in this study, exhibit a differential response to antifungal drugs depending on fungal development stage. Consequently, treating with the MIC of a topical azole antifungal may be effective in mitigating an infection associated with *Aspergillus flavus* in less than 48 h, or *Fusarium* in less than 24 h. However, once the fungal development stage progresses beyond conidial germination when hyphae and mycelium form within the corneal stroma, voriconazole and luliconazole delivered at the MIC as determined by traditional AST becomes less effective at reducing fungal growth and may even contribute to the development of antifungal resistance [28]. Therefore, a better understanding and investigation of the mechanism(s) associated with differential response to azole drugs by fungal conidia and hyphae are warranted. 

One plausible explanation for reduced drug efficacy observed with fungal hyphae is likely related to one of the principal mechanisms of resistance known to operate in filamentous fungi. Plasma membrane multidrug efflux pumps confer resistance to azoles by reducing the intracellular concentration of the drug [28]. These pumps, which are common in species of *Aspergillus* and *Fusarium*, have been shown to undergo upregulation in response to azoles [29]. Additionally, it is not known if the conidial stages of fungi possess these transporters. If they are absent at the conidial stage of development or present and activated post-germination, this may impact on antifungal drug activity against conidia versus hyphae. Further, *Fusarium* efflux pumps are reported to contribute to azole resistance to a greater degree than the efflux pumps found in *Aspergillus* [28], which may in part explain our results of reduced growth inhibition in *Fusarium* cultures treated with an azole at 24 h relative to *Aspergillus*. 

In addition to these proposed mechanisms, Van de Sande et al. (2020) have suggested the cause of reduced antifungal efficacy against mature fungal specimens to be a matter of simple physical access [30]. Fungal hyphae branch and grow at erratic angles forming complex, tortuous networks which may prevent antifungal agents from gaining access to much of the mycelial mass. Van de Sande determined that fungal inocula composed of a hyphal “clump” required a significantly greater inhibitory concentration compared to a homogenous suspension of hyphal fragments [30]. These results are resounding of those obtained from performing antifungal susceptibility testing on biofilms of filamentous fungi. In a study evaluating antifungal activity against in vitro biofilms of *A. fumigatus*, Mowat et al. (2008) demonstrated markedly reduced activity of voriconazole, amphotericin B, and caspofungin against cultures of 24 h growth [17]. Zhang et al. (2012) identified biofilm formation in three keratitis-associated fungal isolates and performed antifungal susceptibility testing on cultured biofilms, revealing a time-dependent decrease in susceptibility against six antifungals [24]. Proposed mechanisms by which biofilms confer antifungal resistance include diffusion of applied antifungal drug, upregulation of efflux pumps, and the presence of extracellular DNA (eDNA), though the exact mechanism by which eDNA confers resistance is yet unknown [16,31,32].

We propose using the models discussed here to perform susceptibility assays which target more advanced stages of fungal development, reflective of morphologies observed in clinical FK. While the classic in vitro AST method has advantages, such as high-throughput screening of high isolate numbers, the ex vivo model described here allows for better replication of the clinical scenario. Aside from being able to evaluate fungal response in the clinically relevant environment of the stroma, this model allows for the aqueous environment of the eye to be refreshed regularly and for fresh application of antifungal drug daily over the course of the assay. This model also has the unique advantage of being able to investigate non-pharmaceutical treatment methods, such as photodynamic light therapy [33]. To improve upon the limitations of this study, we plan to expand the fungal species investigated to include *A. fumigatus* and *F. falciforme*, investigating additional therapies including but not limited to an extended selection and concentrations of antifungals, and use of additional diagnostics such as XTT-measured viability of biofilm structures in vitro, and high-resolution imaging by optical coherence tomography and histology of inoculated cadaver corneas. 

In conclusion, the pursuit of novel, improved therapies for fungal keratitis requires antifungal susceptibility testing that better predicts clinical efficacy. The experimental approaches and model presented in this study will be useful for understanding the interaction between antifungal agents and fungi across different stages of development, especially in the cornea. Our goal is that this improved understanding will be used to not only better inform clinicians on predicated antifungal efficacy but also inform research into developing novel methods of enhancing antifungal drug activity. 

## Figures and Tables

**Figure 1 jof-09-00966-f001:**
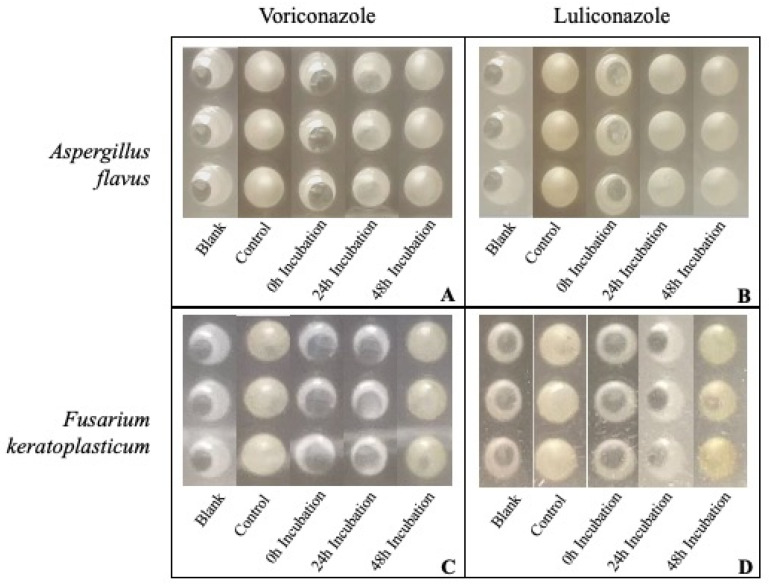
Images of in vitro antifungal susceptibility wells at assay completion (72 h). *Aspergillus flavus* treated with 0.5 μg/mL voriconazole (**A**) or 0.1 μg/mL luliconazole (**B**); *Fusarium keratoplasticum* treated with 8 μg/mL voriconazole (**C**) or 0.2 μg/mL luliconazole (**D**). Wells received antifungal drug after 0, 24, or 48 h incubation post-inoculation. Blank wells were not inoculated; control wells received no antifungal drug.

**Figure 2 jof-09-00966-f002:**
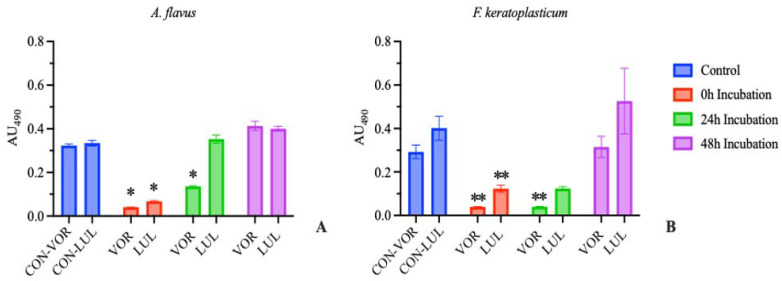
Absorbance at 490 nm at 72 h (mean ± SEM; n = 3). Blank absorbance (mean, n = 3) subtracted from all wells. (**A**) *A. flavus* treatment groups received 0.5 μg/mL voriconazole (VOR) or 0.1 μg/mL luliconazole (LUL). (**B**) *F. keratoplasticum* treatment groups received 8 μg/mL VOR or 0.2 μg/mL LUL. Treatments were initiated after 0, 24, or 48 h incubation. Control groups received no antifungal treatment. Significant differences from control are indicated (* *p* < 0.0005, ** *p* < 0.01).

**Figure 3 jof-09-00966-f003:**
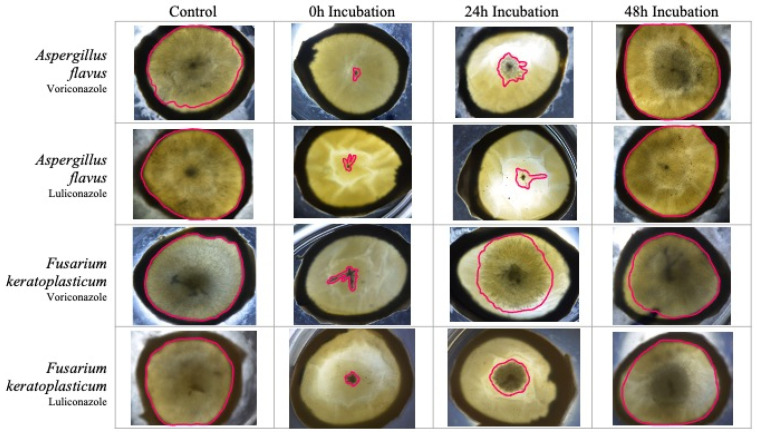
Representative images from each ex vivo corneal treatment group, taken at 72 h post-inoculation. Area of fungal growth outlined in magenta.

**Figure 4 jof-09-00966-f004:**
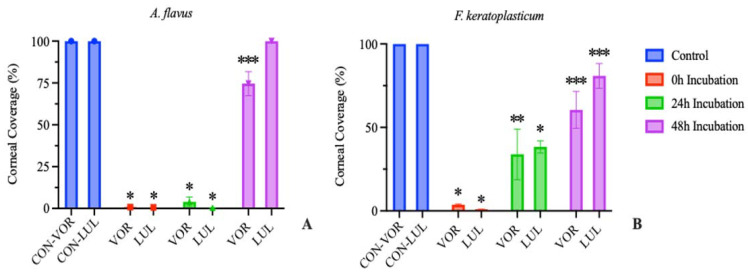
Percentage of total corneal surface area obscured by fungal growth at 72 h post inoculation (mean ± SEM; n = 3). (**A**) *A. flavus* treatment groups received 0.5 μg/mL voriconazole (VOR) or 0.1 μg/mL luliconazole (LUL). (**B**) *F. keratoplasticum* treatment groups received 8 μg/mL VOR or 0.2 μg/mL LUL. Treatments were initiated after 0, 24, or 48 h incubation. Control groups received no antifungal treatment. Significant differences from control are indicated (* *p* < 0.0005, ** *p* < 0.01, *** *p* < 0.02).

## Data Availability

The data presented in this study are openly available in Dryad at https://doi.org/10.5061/dryad.bk3j9kdjf.

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
