# Peer review of "Phase-Dependent Differential In Vitro and Ex Vivo Susceptibility of Aspergillus flavus and Fusarium keratoplasticum to Azole Antifungals"

_jof, 2023, doi:10.3390/jof9100966_

Round 1

Reviewer 1 Report

The author performed research to study phase-dependent differential in vitro and ex vivo susceptibility of Aspergillus flavus and Fusarium keratoplasticum to azole antifungals. The article is well-written and interesting. Here are some recommendations for the author.

1. The genus and species names of microorganisms should be italicized. 

2. The author should provide phase-dependent images, including control, 0, 24 hours, and 48 hours, for the in vitro model.

3. Similarly, phase-dependent images for the in vivo model.

Author Response

Thank you for your considerate review of this manuscript. Please find the detailed responses below and corresponding revisions marked in red in the re-submitted files.

Comment 1. The genus and species names of microorganisms should be italicized.

Response 1: Thank you for pointing this out. All instances of genus and species names have been corrected with italics.

Comment 2. The author should provide phase-dependent images, including control, 0, 24 hours, and 48 hours, for the in vitro model.

Response 2: Thank you; we agree with this suggestion. Accordingly, images of the in vitro model including control, 0, 24, and 48 hour incubation groups have been included. Please see Figure 1, page 5, lines 211-216.

Comment 3. Similarly, phase-dependent images for the in vivo model.

Response 3: We did not perform an in vivo model.  However,  please see Figure 3 (page 6, lines 241-243) for representative images of the corneal ex vivo model, including control, 0, 24, and 48 hour incubation groups.

Reviewer 2 Report

The manuscript jof-2603580 is attractive because characterizes differences in antifungal effect according to phases of fungal development and the authors designed two protocols for its studies. Some considerations are suggested below:

Lines 31, 53, 88, 102, 285, 290, 295, 299, 310, 322, 326, 327, 328, 329, 349: Write the name of fungal species in italics.
Lines 56-67: The mechanism of action of azoles is the inhibition of ergosterol synthesis and these drugs also acts in hyphae, not only in conidia. The MIC assays use conidia because it is not possible to quantify hyphae and padronize a starting inoculum. I suggest that the authors rewrite this paragraph. Some studies hypothesized that fungal growth during human infections could form a biofilm structure and this fact can explain the presence of hyphae in tissues.
Line 136: Add the origin of luliconazole.
Line 145: Why did the authors use the M27 protocol (Reference Method for Broth Dilution Antifungal Susceptibility Testing of Yeasts) and not the M28  document (Reference Method for Broth Dilution Antifungal Susceptibility Testing of Filamentous Fungi) to base the susceptibility tests?
Please, include the CLSI protocol in the reference list.
Line 207: I suggest that the authors add representative images from each time of germination.
Line 253: I also suggest that the authors add representative images from each corneal surface area. Besides, scanning electron microscopy analysis from these samples could reveal more details from each treatment and if the fungal structure is a biofilm after 24h or 48h.
Line 266: Conidia are used to perform antifungal susceptibility tests because it is not possible to quantify hyphae.
Line 304: Write in vitro and ex vivo in italic.
Line 330-337: Another explanation could be the presence of a fungal biofilm.

Author Response

Thank you for your considerate review of this manuscript. Please find the detailed responses below and corresponding revisions marked in red in the re-submitted files.

Comment 1. Lines 31, 53, 88, 102, 285, 290, 295, 299, 310, 322, 326, 327, 328, 329, 349: Write the name of fungal species in italics.

Response 1: Thank you for pointing this out. All instances of fungal species have been corrected with italics.

Comment 2. Lines 56-67: The mechanism of action of azoles is the inhibition of ergosterol synthesis and these drugs also acts in hyphae, not only in conidia. The MIC assays use conidia because it is not possible to quantify hyphae and padronize a starting inoculum. I suggest that the authors rewrite this paragraph. Some studies hypothesized that fungal growth during human infections could form a biofilm structure and this fact can explain the presence of hyphae in tissues.

Response 2: Thank you for the constructive input. We agree with the suggestion made and have rewritten this paragraph, accordingly, removing the discussion on azole mechanism of action and adding a discussion on biofilm formation. Please see lines 56-70 on page 2 for changes.

Comment 3. Line 136: Add the origin of luliconazole.

Response 3: Luliconazole origin (Millipore Sigma, Burlington, MA) added to lines 131-132.

Comment 4. Line 145: Why did the authors use the M27 protocol (Reference Method for Broth Dilution Antifungal Susceptibility Testing of Yeasts) and not the M28  document (Reference Method for Broth Dilution Antifungal Susceptibility Testing of Filamentous Fungi) to base the susceptibility tests? Please, include the CLSI protocol in the reference list.

Response 4: Thank you for pointing this out; our methodology is indeed a modification of the M38 3rd edition, “Reference standard for broth dilution antifungal susceptibility testing of filamentous fungi.” Please see lines 67-70, 137, 286 for corrected in-text references. The appropriate citation has been included in the reference list (line 433).

Comment 5. Line 207: I suggest that the authors add representative images from each time of germination.

Response 5: We agree with this suggestion. Representative germination images have been included as a supplemental figure.

Comment 6. Line 253: I also suggest that the authors add representative images from each corneal surface area. Besides, scanning electron microscopy analysis from these samples could reveal more details from each treatment and if the fungal structure is a biofilm after 24h or 48h.

Response 6: Thank you for this suggestion. We have added representative images of corneal surface areas from the ex vivo model, including control, 0, 24, and 48 hour incubation groups. Please see Figure 3 (page 6, lines 241-243). Scanning electron microscopy would only image the surface of the cornea and not the stromal infection which is typical of clinical disease.  We are investigating other imaging techniques such as optical coherence tomography or specific histology which we will explore this technique in future investigations.

Comment 7. Line 266: Conidia are used to perform antifungal susceptibility tests because it is not possible to quantify hyphae.

Response 7: Please see lines 272-284 on page 7 for an updated discussion, recognizing this specific limitation in quantifying a hyphal suspension.  

Comment 8. Line 304: Write in vitro and ex vivo in italic.

Response 8: In vitro” and “ex vivo” have been corrected with italics. 

Comment 9. Line 330-337: Another explanation could be the presence of a fungal biofilm.

Response 9: Agreed. Please see lines 342-352 on page 9 for the addition of a discussion on fungal biofilm formation and contribution to disease.

Reviewer 3 Report

To characterize the response of different stage of fungal development to antifungal treatment is an interesting topic. However, it was not surprise that 24h and 48h hyphae was more resistance than the conidia, since it has formed biofilm at this stage.   

In addition, the measurement of absorbance was not suitable to value the efficiency of antifungal treatment to hyphae. Instead, the metabolic assay (XTT, MTT) was preferred.

Author Response

Thank you for your considerate review of this manuscript. Please find the detailed responses below and corresponding revisions marked in red in the re-submitted files.

Comment 1. To characterize the response of different stage of fungal development to antifungal treatment is an interesting topic. However, it was not surprise that 24h and 48h hyphae was more resistance than the conidia, since it has formed biofilm at this stage.   

Response 1: Thank you for the constructive input. We agree and have adding a discussion on biofilm formation and its possible role in drug resistance. Please see lines 56-70 on page 2 for changes.

Comment 2. In addition, the measurement of absorbance was not suitable to value the efficiency of antifungal treatment to hyphae. Instead, the metabolic assay (XTT, MTT) was preferred.

Response 2: Thank you for this recommendation. We selected the measurement of absorbance as a relative, quantitative diagnostic for use in addition to visual assessment with a magnified reading mirror, as is described in the CLSI M38 reference standard. We agree with the recommendation to consider the inclusion of a metabolic assay and will include this technique in future investigations. Please see line 365 for a discussion on the inclusion of this assay in future work.

Round 2

Reviewer 1 Report

The author has replied well to my comment.

Reviewer 3 Report

The authors have addressed all my concerns. No more questions.